# Unified metal-free intermolecular Heck-type sulfonylation, cyanation, amination, amidation of alkenes by thianthrenation

Ming-Shang Liu[1], Hai-Wu Du[1,2], Huan Meng[1], Ying Xie[3] & Wei Shu [1,2,3] ✉

Direct and site-selective C-H functionalization of alkenes under environmentally benign conditions represents a useful and attractive yet challenging transformation to access value-added molecules. Herein, a unified protocol for a variety of intermolecular Heck-type functionalizations of $C_{sp2}$-H bond of alkenes has been developed by thianthrenation. The reaction features metal-free and operationally simple conditions for exclusive *cine*-selective C-H functionalization of aliphatic and aryl alkenes to forge C-C, C-N, C-P, and C-S bonds at room temperature, providing a general protocol for intermolecular Heck-type reaction of alkenes with nucleophiles (Nu = sulfinates, cyanides, amines, amides). Alkenes undergo *cine*-sulfonylation, cyanation, amination to afford alkenyl sulfones, alkenyl nitriles and enamines.

Alkenes represent one of the most useful functional groups due to their profound potential to a myriad of other functional groups as well their orthogonal reactivity over other polar functional groups[1–4]. Owing to the abundance, diversity, and easy-availability of alkenes, developing efficient and practical functionalizations of alkenes has been a long-term preoccupation in synthetic chemistry[5,6]. Among which, Heck reaction is one of the most straightforward and efficient means to functionalize alkenes[7–9]. Typically, Heck reaction gives access to *ipso*-substitution of alkenes where a leaving group is bonded to the olefinic carbon atom by nucleophilic species. Comparably, *cine*-substitution of Heck reaction are less investigated. Over the past decades, transition-metal-catalyzed Heck-type *cine*-arylation and vinylation of electron-deficient alkenes bearing a leaving group have been developed[10–18]. Straightforward and environmentally benign Heck-type methods that transform alkenes into versatile carbon electrophiles would be highly desirable[19,20]. In particular, alkenyl sulfones, alkenyl nitriles, enamines, and enamides are of importance in pharmaceuticals, biochemistry, and materials sciences[21–24], providing a straightfoward opportunity in multistep organic synthesis or covalent modification of proteins in drug discovery to couple with different electrophiles. Therefore, direct and regioselective sulfonylation, cyanation, amination, amidation of alkenes would be an enabling synthetic tool to access such privileged structures. To date, metal-free

intermolecular Heck-type reaction of alkenes to access alkenyl sulfones[25], alkenyl nitriles[26], enamines, and enamides remains underdeveloped[27,28].

On the other hand, alkenyl thianthrenium salts[29–40] have been considered as one umpolung strategy of alkenes for further chemical synthesis pioneered by Shine[41,42]. Recently, Ritter developed the practical and scalable synthesis of alkenyl thianthrenium salts[43,44], creating new opportunities for derivatization of unactivated alkenes[15,45,46]. In particular, metal-free functionalization of alkenes represent an attractive aspect to functionalize alkenes under mild conditions. In 2021, elegant examples of electrochemical aziridination of alkenes with primary amines have been demonstrated with or without thianthrene[47,48]. The generation of dicationic intermediates also offers potential opportunities for *ipso*- and *cine*-substitution reactions. In 2022, Shu group developed a unified metal-free intermolecular aziridination and cyclopropanation of alkenes by thianthrenation (Fig. 1b, top)[49]. Sulfonamides, carbamates, amides, primary amines, and methylenes with acidic protons were all successfully employed as nucleophiles. In 2021, Wickens and Shu independently reported the allylic functionalizations of alkenes by thianthrenation to from C–N, C–C, C–O, and C–S bonds in the presence of nucleophiles (Fig. 1b, middle)[50,51]. Recently, Soós group developed an ene-type Kornblum-Ganem oxidation of alkenes by thianthrenation to access

[1]Shenzhen Grubbs Institute and Department of Chemistry, Guangming Advanced Research Institute, Southern University of Science and Technology, 518055 Shenzhen, Guangdong, P. R. China. [2]State Key Laboratory of Elemento-Organic Chemistry, Nankai University, 300071 Tianjin, P. R. China. [3]College of Chemistry and Environmental Engineering, Sichuan University of Science and Engineering, 643000 Zigong, P. R. China. ✉e-mail: shuw@sustech.edu.cn

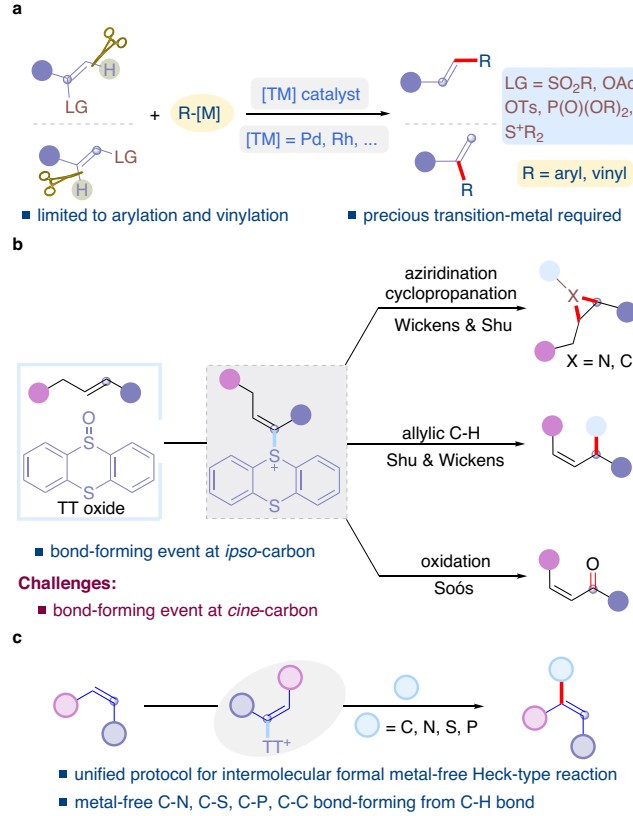

a

■ limited to arylation and vinylation
■ precious transition-metal required

LG = SO₂R, OAc, OTs, P(O)(OR)₂, S⁺R₂

R = aryl, vinyl

b

aziridination cyclopropanation
Wickens & Shu

X = N, C

allylic C-H
Shu & Wickens

oxidation
Soós

TT oxide

■ bond-forming event at *ipso*-carbon

**Challenges:**

■ bond-forming event at *cine*-carbon

c

■ unified protocol for intermolecular formal metal-free Heck-type reaction
■ metal-free C-N, C-S, C-P, C-C bond-forming from C-H bond
■ internal & terminal, aryl & aliphatic alkenes
■ mild conditions and broad scope

**Fig. 1 | Impetus for metal-free functionalizations of alkenes by thianthrenation.**
**a** Transition-metal-catalyzed *cine*-substitution of alkene electrophiles.
**b** Representative metal-free functionalization modes of alkenes by thianthrenation.
**c** Metal-free *cine*-functionalizations of alkenes by thianthrenation (this work).

various α,β-unsaturated carbonyls (Fig. 1b, bottom)[52]. Interestingly, metal-free transformations of alkenes by thianthrenation basically led to bond-formation at *ipso*-carbon of alkenyl thianthrenium salts. We questioned the possibility of realizing a new bond-formation mode of alkenyl thianthrenium salts to functionalization *cine*-carbon of alkenes under metal-free conditions[48]. Herein, we report a unified protocol for metal-free *cine*-functionalizations of alkenes by thianthrenation (Fig. 1c). The reaction explores the new reactivity of alkenyl thianthrenium salts to form a new chemical bond at *cine*-position instead of *ipso*-position of vinyl thianthrenium salts. The mild condition allows for the site-selective C–H functionalization of alkenes to forge C–S, C–N, C–P and C–C bonds with diverse nucleophiles.

## Results

### Optimization of the reaction conditions

We started the investigation using sodium methanesulfinate (**1a**) and 4-phenylbut-1-enylthianthrenium salt (**2a**) as model substrates to evaluate the reaction conditions (Table 1). To our delight, *cine*-sulfonation of the C = C bond of the vinyl thianthrenium salt was exclusively formed, without the formation of formal allylic C-H sulfonation byproduct (**3a'**) as previously reported. After evaluation of the reaction parameters, we define the reaction in DCE (0.1 M) at room temperature without any additive as standard conditions, providing the desired product (3-(methylsulfonyl)but-3-en-1-yl)benzene **3a** in 83% isolated yield (Table 1, entry 1). The use of other solvents instead of DCE could also mediate the desired transformation, albeit giving **3a** in lower yields (Table 1, entries 2–9).

### Scope of the reaction

With the optimized conditions in hand, the scope of alkenes and sodium sulfinates is examined and the results are summarized in Fig. 2. First, the scope of alkenes was evaluated. A wide range of alkenes with diverse electronic and steric properties are suitable for this reaction, allowing the corresponding *cine*-sulfonylation of alkenes by thianthrenation with sodium methanesulfinate in good yields (**3a-3z**). Aliphatic terminal alkene-based thianthrenium salts are all compatible

---

## Table 1 | Condition evaluation for the *cine*-sulfonation.[a]

MeSO₂Na + Ph〜〜TT⁺ BF₄⁻ → (DCE (0.1 M), rt, 10 h) → **3a** (Ph〜〜SO₂Me) / **3a'** (Ph〜〜SO₂Me)
**1a**        **2a**

| Entry | Variations as shown | Conversion of 2a | yield of 3a |
|---|---|---|---|
| 1 | None | >95% | 88% (83%)[b] |
| 2 | DCM instead of DCE | >95% | 78% |
| 3 | CH₃CN instead of DCE | 90% | 56% |
| 4 | THF instead of DCE | >95% | 75% |
| 5 | DME instead of DCE | >95% | 77% |
| 6 | toluene instead of DCE | >95% | 76% |
| 7 | DMA instead of DCE | >95% | 52% |
| 8 | DMF instead of DCE | >95% | 63% |
| 9 | DMSO instead of DCE | >95% | 65% |

[a]The reaction was conducted using **1a** (0.15 mmol) and **2a** (0.10 mmol) under indicated conditions at room temperature. Yield was determined by ¹H NMR of the crude mixture using mesitylene as the internal standard.
[b]Isolated yield after flash chromatography.

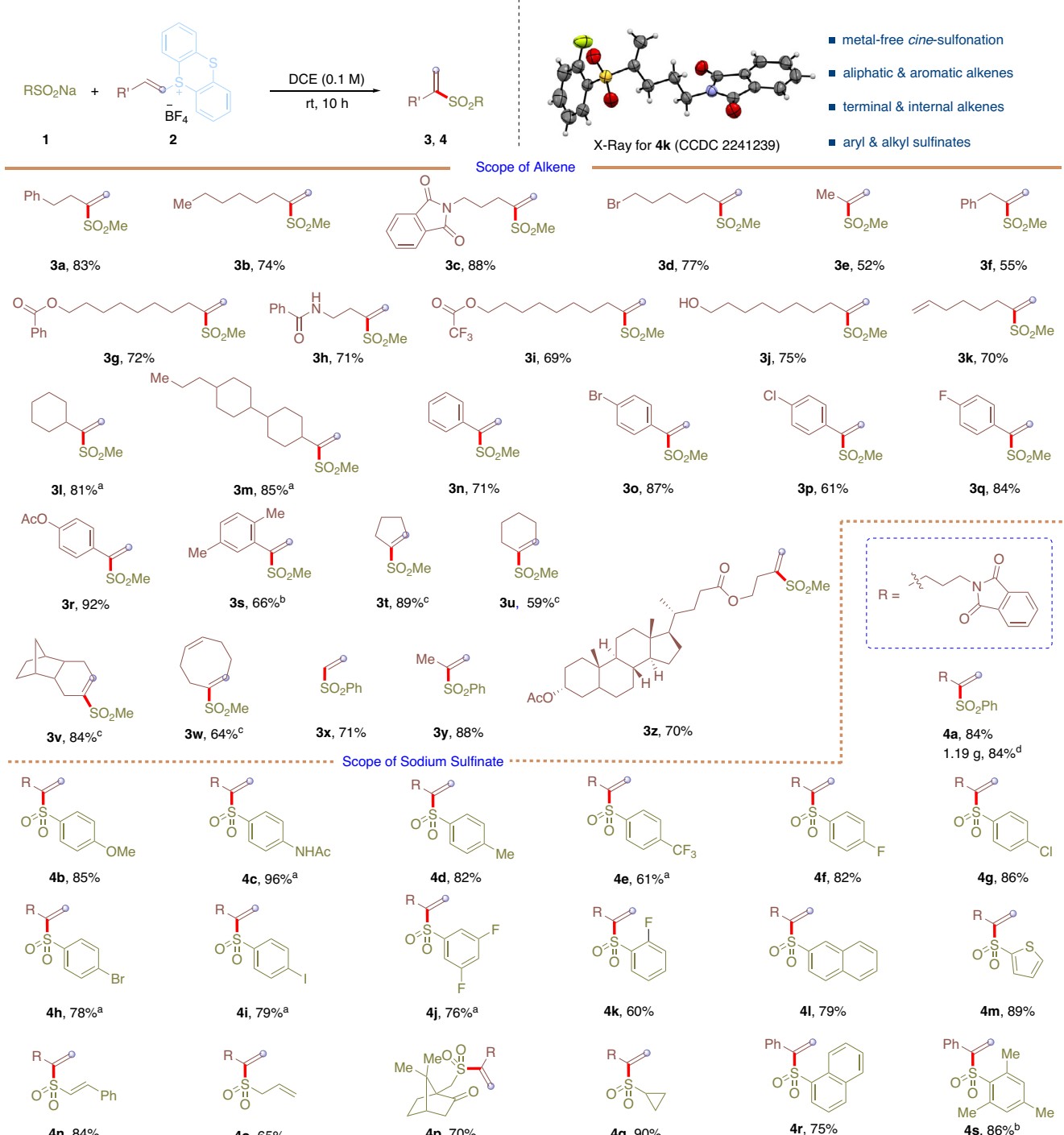

**Fig. 2 | Scope of *cine*-sulfonylation of alkenes by thianthrenation.** Standard conditions, see Table 1 for details. Isolated yield is shown. [a]Reaction was run for 24 h. [b]Reaction was conducted at 50 °C. [c]Reaction was run at 50 °C for 36 h. [d]Reaction was conducted on 4.0 mmol scale.

in this reaction, producing alkenyl sulfones in 52–88% yields (**3a-3m**). Alkenes with pendant amides, bromides, esters were compatible in the reaction, giving the corresponding *cine*-substitution products (**3c, 3d, 3g-3i**) in 69–88% yields. It is noteworthy that amides with free N–H, free alcohols, and alkenes were all compatible in the reaction to furnish the desired alkenyl sulfones (**3h, 3j**, and **3k**) in 70–75% yields, leaving chemical space for further elaboration. In addition, α-branched aliphatic alkene-based thianthrenium salts are also good substrates in the reaction, giving corresponding *cine*-sulfonylation products in 81% and 85% yields (**3l** and **3m**). Moreover, styrenes could be efficiently involved in the *cine*-substitution process by thianthrenation, giving the

desired sulfones in 61–92% yields (**3n-3s**). In addition, cyclic alkenes were easily converted to alkenylsulfones in 59–89% yields (**3t-3w**) under the reaction conditions. Notably, gaseous alkenes, such as ethylene and propene, could be successfully involved in this *cine*-substitution process by thianthrenation, giving **3e, 3x** and **3y** in 52–88% yields. Lithocholic acid drived alkene with molecular complexity underwent *cine*-substitution process smoothly, giving the desired product (**3z**) in 70% yield.

Next, the scope of sulfinates was tested. *para*-Substituted aryl sulfinates with electron-donating (**4a-4d**) or electron-withdrawing (**4e-4i**) groups were all well-tolerated in this reaction, giving

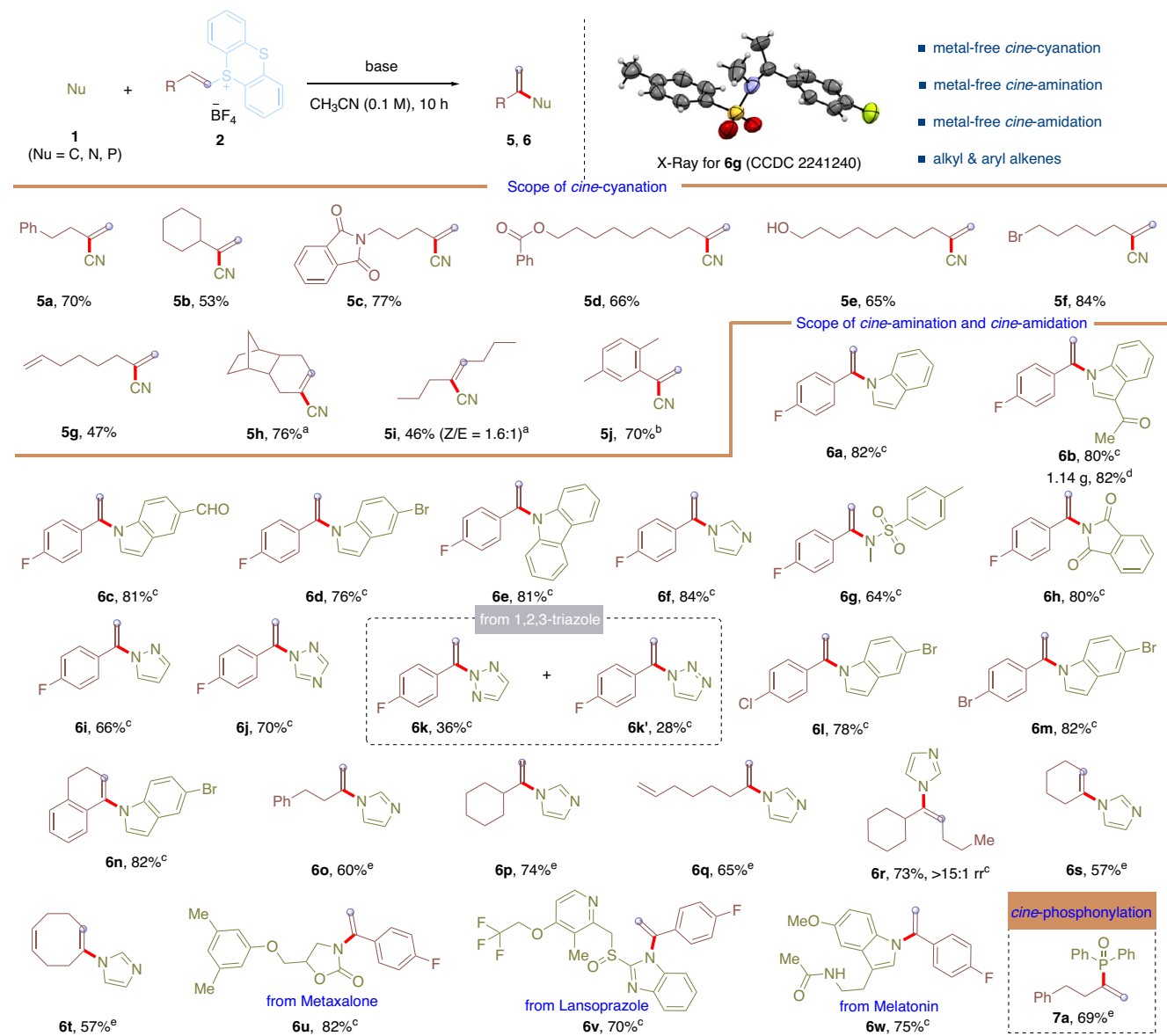

**Fig. 3 | Scope of the *cine*-cyanation, *cine*-amination and *cine*-amidation of alkenes by thianthrenation.** The reaction was carried out using **1** (0.15 mmol), **2** (0.10 mmol), KF (3.0 equiv) in CH₃CN (0.1 M) at room temperature for 10 h unless otherwise stated. Isolated yield is shown. ᵃThe reaction was conducted at 50 °C.

ᵇThe reaction was conducted at 50 °C for 24 h. ᶜThe reaction was conducted using K₂CO₃ (1.0 equiv). rr = ratio of regioisomers. ᵈThe reaction was conducted on 5.0 mmol scale using K₂CO₃ (1.0 equiv). ᵉThe reaction was conducted using Cs₂CO₃ (1.0 equiv).

corresponding *cine*-sulfonylation products in good yields (61–96%). Moreover, *meta*- and *ortho*-substituted aryl sulfinates were also good substrates for this reaction to give the desired products (**4j** and **4k**) in 76% and 60% yields, respectively. Fused aryl and heteroaryl sulfinates underwent *cine*-substitution to give the desired products (**4l** and **4m**) in 79% and 89% yields. Vinyl sulfinate proceeded smoothly to give the corresponding alkenyl sulfone **4n** in 84% yield. Notably, allylic, acyclic and cyclic alkyl sulfinates were compatible in this metal-free *cine*-substitution process and afforded the corresponding sulfones in 65-90% yields (**4o-4q**). It is noteworthy that bulky sulfinates smoothly underwent *cine*-substitution of C-H bond by thianthrenation to give the corresponding alkenyl sulfones in 75% and 86% yields (**4r** and **4s**). In addition, the structure of the alkenyl sulfones was unambiguously confirmed by X-ray diffraction of **4k**. The reaction could be scaled up to 4.0 mmol to afford **4a** in 84% yield (1.16 g), rendering the reaction useful for large-scale synthesis.

Furthermore, the protocol for *cine*-functionalizations of alkenes by thianthrenation was further applied to *cine*-cyanation, *cine*-

amination, and *cine*-amidation to forge C-C bonds and C-N bonds from C-H bonds (Fig. 3). With slight modification of the solvent and base, *cine*-cyanation was achieved with zinc cyanide in the presence of KF (3.0 equiv) in CH₃CN (0.1 M) at room temperature, delivering the desired 2-methylene-4-phenylbutanenitrile (**5a**) in 70% isolated yield (Supplementary Tables 2 and 3) (for details see Supplementary Information)[48]. Then, the scope of *cine*-cyanation was evaluated. α-Branched terminal alkene derived thianthrenium salt is compatible with this *cine*-cyanation, delivering the corresponding *cine*-substitution product (**5b**) in 53% yield. Additionally, aliphatic alkenes tethered with esters, amides, alcohols were all compatible in this reaction, affording the desired acrylonitriles in 65-77% yields (**5c-5e**). Moreover, alkenes with bromides and alkenes were successfully transformed to the *cine*-cyanation products in 84% and 47% yields (**5f** and **5g**). Notably, cyclic and acyclic internal aliphatic alkenes derived thianthrenium salts were amenable to this *cine*-cyanation, affording diverse acrylonitriles in 76% and 46% yields (**5h** and **5i**). Interestingly, styrenes also worked well for this *cine*-cyanation to afford **5j** in 70% yield.

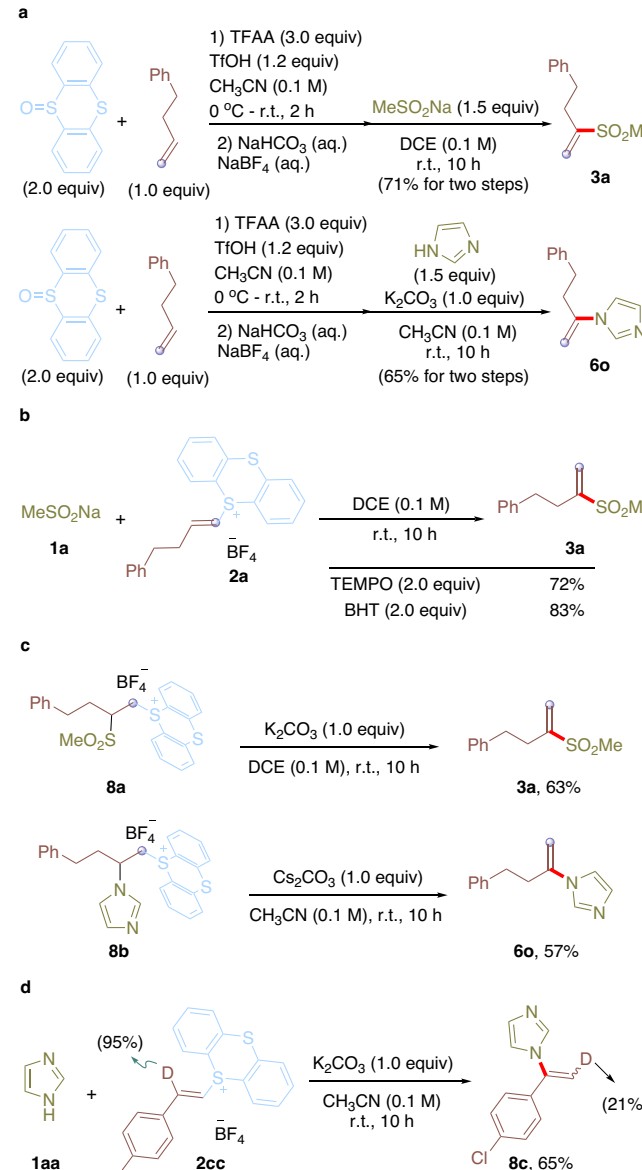

**Fig. 4 | One-pot *cine*-functionalizations of alkenes and mechanistic investigations. a** One-pot reactions for *cine*-substitution of alkenes. **b** Probe radical nature of the reaction. **c** Assign reaction intermediates. **d** Reaction with deuterated vinyl thianthrenium salt.

Impressively, this operationally simple *cine*-substitution protocol could be successfully applied to *cine*-amination of C-H bonds using nitrogen nucleophiles (Fig. 3). The reaction of **2q** with indole in the presence of K₂CO₃ (1.0 equiv) in CH₃CN (0.1 M) at room temperature afforded the selective *cine*-substitution at nitrogen product 1-(1-(4-fluorophenyl)vinyl)-1*H*-indole **6a** in 82% isolated yield (Supplementary Tables 4 and 5) (for details see Supplementary Information). Ketone-, aldehyde-, and bromo-substituted indoles were well-tolerated under this *cine*-amination conditions, delivering the corresponding *N*-vinyl indoles (**6b**-**6d**) in 76 − 81% yields. Additionally, carbazole and imidazole were all excellent substrates for this *cine*-substitution, yielding the desired *N*-vinyl carbazole (**6e**) and imidazole (**6f**) in 81% and 84% yields. Moreover, sulfonamides and amides were smoothly transformed into corresponding *N*-vinyl amides in 64% and 80% yields (**6g** and **6h**). Furthermore, pyrazoles and 1,2,4-triazoles were remarkable substrates for this *cine*-substitution reaction, producing the desired *N*-vinyl pyrazole (**6i**) and triazole (**6j**) in 66% and 70% yields. 1,2,3-Triazoles were

also compatible in the reaction, delivering a mixture of **6k** and **6k'** in 36% and 28% yields. Moreover, a variety of alkenes worked well the C-N bond-forming process from *cine*-C-H bond of alkenes by thianthrenation. Chloro-, bromo-substituted styrenes and internal styrenes were compatible with this *cine*-substitution reaction, generating *cine*-amination products (**6l**-**6n**) in 78-82% yields. Linear and α-branched aliphatic alkenes derived thianthrenium salts were all tolerated under the *cine*-amination conditions, delivering corresponding enamines (**6o** and **6p**) in 60% and 74% yields. Isolated diene selectively underwent thianthrenation and sequential *cine*-amination on one alkene, giving corresponding product **6q** in 65% yield. Unsymmetrical alkenes could be involved in the regioselective thianthrenation and sequential *cine*-amination to yield **6r** in 73% yield with >15:1 rr. Cyclic alkenes with different ring size could be involved to this *cine*-substitution with imidazole to furnish enamines (**6s** and **6t**) in 57% yield. Additionally, drug molecules have been derivatized. Metaxalone, and lansoprazole underwent selective *cine*-amination reaction of the C-H bond of alkenes with amides and benzoimidazoles to give *N*-acyl and *N*-aryl enamines (**6u** and **6v**) in 82% and 70% yields, respectively. Interestingly, melatonin underwent chemoselective *N*-vinylation with of indoles instead of the amides at *cine*-position of alkenes to give *N*-aryl enamine derivative (**6w**) in 75% yield under standard conditions. Additionally, *cine*-phosphonylation product **7a** was got in 69% yield from phosphoryl nucleophile and alkenylthianthrenium salt.

## Mechanistic study

To enhance the practicality of this operationally simple protocol, a one-pot procedure was evaluated for *cine*-sulfonylation and amination (Fig. 4a). The one pot thianthrenation of 4-phenyl-1-butene using thianthrene *S*-oxide, followed by the reaction of sodium methanesulfinate or imidazole under corresponding conditions were conducted, affording the desired *cine*-substituted products in comparable yields (**3a**, 71%) or (**6o**, 65%) without any intermediate purification. Furthermore, the reactions of **1a** and **2a** in the presence of 2 equiv of radical scavenger (TEMPO or BHT) were carried out under otherwise identical to standard conditions, providing the desired alkenyl sulfone **3a** in 72% and 83% yields (Fig. 4b). The result that the presence of TEMPO or BHT did not decrease the efficiency of this reaction, excluding the radical pathway of this *cine*-substitution reaction. To further detect the mechanism of this *cine*-substitution, the preformed alkyl thianthrenium salts (**8a** and **8b**) were subjected to nucleophiles under standard conditions (Fig. 4c). Corresponding alkenyl sulfone **3a** and alkenyl amine **6o** were obtained in 63% and 57% yield, suggesting the primary alkyl thianthrenium salts may serve as the intermediate for the selective *cine*-substitution process. Additionally, the reaction of **1aa** and deuterated alkenyl thianthrenium salt **2cc** was conducted under standard conditions, affording the desired *cine*-amination product **8c** in 65% yield. Interestingly, partial deuterium scrambling was observed (Fig. 4d), supporting the proton exchange of β-position of alkenyl thianthrenium salt with surroundings and protonation of α-position of alkenyl thianthrenium.

Based on the mechanistic experiments and literature[48–52], a plausible mechanism of the metal-free *cine*-substitution of alkenyl thianthrenium salts is proposed and depicted in Fig. 5. First, the zwitterionic alkyl thianthrenium salt intermediate **M1** could be formed by intermolecular addition of nucleophiles on distal position of alkenes to forge C-S/C-C/C-N/C-P bonds. After protonation of **M1** to form a more stable intermediate **M2**, **M2** underwent deprotonation on *cine*-site yield the zwitterion **M3**. **M3** could undergo intramolecular elimination to afford alkenyl sulfones, acrylonitriles, enamines, and enamides by releasing thianthrene.

## Discussion

In conclusion, a unified metal-free protocol for diverse intermolecular *cine*-functionalizations of the C−H bond of alkenes by thianthrenation

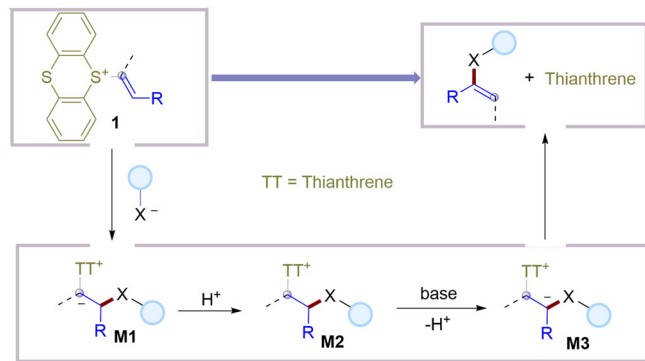

**Fig. 5 | Proposed mechanism of the reaction.** The plausible mechanism for metal-free *cine*-functionalization of alkenes by thianthrenation.

has been achieved. The reaction features metal-free C–H functionalizations of alkenes under mild conditions to forge C–S, C–C, C–N, and C–P bonds from C–H bonds via *cine*-sulfonylation, *cine*-cyanation, *cine*-amination, *cine*-amidation, and *cine*-phosphonylation. The reaction represents new metal-free reaction mode to functionalize *cine*-carbon of alkenyl thianthrenium salts, which is complementary to previous functionalization at *ipso*-carbon. Mechanistic investigations revealed the reaction undergo site-selective nucleophilic addition followed by regioselective elimination to afford the formal Heck reaction of alkenes, affording synthetic useful synthons which are difficult to access from readily accessible starting materials.

## Methods

### General procedure A for intermolecular Heck-type sulfonylation of alkenes by thianthrenation
Sodium sulfinate (0.15 mmol) and vinyl thianthrenium salt (0.1 mmol) were placed in a 10.0 mL Schlenk tube which equipped with a magnetic stir bar. After back-filled with nitrogen (this process was repeated three times), DCE (1.0 mL) was added. The vial was sealed and at room temperature (for the large hindrance substrates its require at 50 °C) with stirring until TLC indicated the complete consumption of thianthrene (typically 10 h or 36 h). The reaction mixture was evaporated and purified directly by column chromatography to afford the product.

### General procedure B for intermolecular Heck-type cyanation of alkenes by thianthrenation
$Zn(CN)_2$ (17.6 mg), KF (17.4 mg) and vinyl thianthrenium salt (0.1 mmol) were placed in a 10.0 mL Schlenk tube which equipped with a magnetic stir bar. After back-filled with nitrogen (this process was repeated three times), $CH_3CN$ (1.0 mL) was added. The vial was sealed and at room temperature (for the large hindrance substrates it require at 50 °C) with stirring until TLC indicated the complete consumption of thianthrene (typically 10 h or 24 h). The reaction mixture was evaporated and purified directly by column chromatography to afford the product.

### General procedure C for intermolecular Heck-type amination and amidation of styrenes by thianthrenation
Nucleophile (0.15 mmol), $K_2CO_3$ (13.8 mg) and vinyl thianthrenium salt (0.1 mmol) were placed in a 10.0 mL Schlenk tube which equipped with a magnetic stir bar. After back-filled with nitrogen (this process was repeated three times), $CH_3CN$ (1.0 mL) was added. The vial was sealed and at room temperature with stirring until TLC indicated the complete consumption of thianthrene (typically 10 h). The reaction mixture was evaporated and purified directly by column chromatography to afford the product.

### General procedure D for intermolecular Heck-type amination and amidation of aliphatic alkenes by thianthrenation
Nucleophile (0.15 mmol), $Cs_2CO_3$ (32.6 mg) and vinyl thianthrenium salt (0.1 mmol) were placed in a 10.0 mL Schlenk tube which equipped with a magnetic stir bar. After back-filled with nitrogen (this process was repeated three times), $CH_3CN$ (1.0 mL) was added. The vial was sealed and at room temperature with stirring until TLC indicated the complete consumption of thianthrene (typically 10 h). The reaction mixture was evaporated and purified directly by column chromatography to afford the product.

## Data availability
The X-ray crystallographic coordinates for structures that support the findings of this study have been deposited at the Cambridge Crystallographic Data Center (CCDC) with the accession codes CCDC 2241239 (**4k**) and CCDC 2241240 (**6g**) via www.ccdc.cam.ac.uk/data_request/cif. The authors declare that all other data supporting the findings of this study are available within the article and Supplementary Information files, and also are available from the corresponding author upon request.

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

## Acknowledgements

Financial support from NSFC (21971101, 22171127, 22371115), Guangdong Basic and Applied Basic Research Foundation (2022A1515011806), Department of Education of Guangdong Province (2021KTSCX106, 2022JGXM054), Shenzhen Science and Technology Innovation Com-mittee (JCYJ20220530114606013, JCYJ20230807093522044), The Pearl River Talent Recruitment Program (2019QN01Y261), Guangdong Provincial Key Laboratory of Catalysis (No. 2020B121201002) is sincerely acknowledged. We thank Prof. Wickens (UWM) for insightful sugges-tions and comments. We acknowledge the assistance of SUSTech Core Research Facilities. We thank Dr. Xiaoyong Chang (SUSTech) for assis-tance with the X-ray crystallographic analysis of **4k** (CCDC 2241239), **6g** (CCDC 2241240), and Dr. Quan-Xing Zi (SUSTech) for reproducing the results of **3h**, **4k**, **5g** and **6m**.

## Author contributions

M.S.L. discovered and developed the reaction. W.S. conceived and directed the project. M.S.L. performed the experiments, M.S.L. and

H.W.D. collected the data. M.S.L. and H.W.D. synthesized the substrate materials. H.M. and Y.X. discussed the project with W.S. and helped prepare the manuscript. All authors discussed and analyzed the data. W.S and M.S.L. wrote the manuscript with contributions from other authors.

## Competing interests

The authors declare no competing interests.
