## [Peer Review File · Nature Communications]

REVIEWER COMMENTS

Reviewer #1 (Remarks to the Author):

In this manuscript, Shu and co-workers reported a unified metal-free protocol for diverse intermolecular cine-functionalizations of the C-H bond of alkenes by thianthrenation. This transformation features metal-free and operationally simple conditions for exclusive cine-selective C-H functionalization of aliphatic and aryl alkenes to forge C-C, C-N, and C-S bonds at room temperature, providing a general protocol for intermolecular Heck-type reaction of alkenes with nucleophiles (Nu = sulfinates, cyanides, amines, amides). Alkenes undergo cine-sulfonylation, cyanation, amination to afford alkenyl sulfones, alkenyl nitriles and enamines. This work has not only revealed diverse and unique reactivity from alkenyl sulfonium salts but also broadened the reaction scope and caused the related chemistry to flourish. Overall, I would recommend publication of the manuscript in Nature Communication after some minor comments are addressed.

1. "Table 1. Condition evaluation for the amides synthesis from aldehydes and imines". This sentence doesn't agree with the following content. Please revised it.
2. Whether other nucleophilic reagents such as oxygen or sulfur nucleophilic reagents can also undergo cine substitution?
3. For the cine-amination, have the authors tried other sources of amines such as pyrazole, or 1,2,3-triazole, and what's the result?

Reviewer #2 (Remarks to the Author):

Recommendation: Publish in Nature Communications after minor revisions.

Comments:

In this manuscript, Shu and coworkers have developed a cine-substitution reaction of alkenyl thianthrenium salts with nucleophiles (Nu = sulfinates, cyanides, amines, amides) to access alkenyl sulfones, alkenyl nitriles, enamines and enamides. The cine-substitution includes cine-sulfonylation,

cine-cyanation, cine-amination, and cine-amidation via C-C, C-N, and C-S bond formation under the mild conditions. The thianthrenium salts can be easily prepared from olefins as demonstrated in the references. This protocol can also be expanded to use the alkenes as the starting materials via the intermediacy of thianthrenium salts in one-pot. This method offers an advantage and is interesting to synthetic chemists. Therefore, the manuscript is recommended for publication in Nature Communications.

Nevertheless, minor revision is recommended considering following questions.

1. Page 4-5: X-ray for 4j (CCDC 2241239) is shown. However, from the picture of the structure, it seems that it is 4k rather than 4j. Please check it.
2. Reference (Org. Lett. 2023, 25, 5324–5328) is suggested to be added for the description of “On the other hand, alkenyl thianthrenium salts²⁹⁻³⁹...” (page 2)
3. Page 9: The sentence “...the formal Heck reaction of alkenes. access to synthetic useful synthons which are difficult to access from readily accessible starting materials” should be revised.
4. Is this reaction specific for cine-substitution or selective for cine-substitution? Is there any ipso-substitution product observed particularly when the reaction yield is not high for cine-substitution product?
5. Page 8: Regarding the mechanism, is there any evidence for the proton transfer and elimination as proposed?

Reviewer #3 (Remarks to the Author):

Comments about NCOMMS-23-34197-T

In this manuscript, the authors report a unified platform for Heck-type functionalization of olefins by thianthrenation. The strategy is simple in process, without the employment of transition metals, and selectively provides products of Heck-type cine-substitution. This protocol offered new ideas for the construction of cine-C-S, cine-C-C and cine-C-N bonds from olefinic C-H bonds, and is a completely

new paradigm in olefinic thianthrene salt chemistry. However, I still have many questions before the article is published. After clearing these confusions, it can be considered for publication in Nature Communications.

1. Are some olefins with electron-withdrawing groups reactive?
2. The crystallographic data in Figure 2 appear to be 4k rather than 4j.
3. The yields of all alkenyl thianthrenium salts should be stated - I note that some of the substrates show yields, but not all.
4. The authors should add examples of asymmetric olefins, e.g. one end of 5i becomes butyl or a longer chain.
5. In the authors' previous work (doi: 10.1039/D1SC06577G), the allylic product was obtained with 85% yield using CS₂CO₃ as base; the same 82% yield was maintained using CH₃CN as solvent. However, in this manuscript, using the same experimental conditions, the Heck-type cine-functionalized product (e.g. product 6l, 6m...) was obtained. I think the solvent effect is the key to dominating both products. Please explain the difference between these two strategies in the Mechanistic Investigation section, otherwise, it is easy to mislead our large readership.
6. The assigned reactions for the intermediates of the reaction are brilliantly designed. But this raises new questions: under the current reaction conditions, C-S⁺ bond breaking can also occur from the stabilized intermediates (7a & 7b). So is it possible that the C-S⁺ bond of alkenyl thianthrenium salts also breaks at the beginning of the reaction and generates an ipso-carbon-negative ion; in such a case the nucleophilic reagent would also attack the cine-position more. Perhaps the authors could experiment with a reaction in which a photocatalyst (or a transition-metal catalyst, but this would probably generate products with only ipso-substitution) is used to induce C-S⁺ bond rupture in the presence of light, generating an alkenyl radical intermediate, and then the structure of products could be observed to corroborate the mechanism recounted in the paper.

Point-to-Point Response to Reviewers' Comments

Reviewer 1: In this manuscript, Shu and co-workers reported a unified metal-free protocol for diverse intermolecular *cine*-functionalizations of the C-H bond of alkenes by thianthrenation. This transformation features metal-free and operationally simple conditions for exclusive *cine*-selective C-H functionalization of aliphatic and aryl alkenes to forge C-C, C-N, and C-S bonds at room temperature, providing a general protocol for intermolecular Heck-type reaction of alkenes with nucleophiles (Nu = sulfinates, cyanides, amines, amides). Alkenes undergo *cine*-sulfonylation, cyanation, amination to afford alkenyl sulfones, alkenyl nitriles and enamines. This work has not only revealed diverse and unique reactivity from alkenyl sulfonium salts but also broadened the reaction scope and caused the related chemistry to flourish. Overall, I would recommend publication of the manuscript in Nature Communication after some minor comments are addressed.

General response: We thank the reviewer for supporting the publish of this work in *Nature Communications*! As suggested, the comments are fully addressed in the following sections in detail and corresponding details are provided in the revised manuscript and Supplementary Information.

Comment 1: “Table 1. Condition evaluation for the amides synthesis from aldehydes and imines”. This sentence doesn't agree with the following content. Please revised it.

Our response: We thank the reviewer for bringing out this point. As suggested, we correct the title of Table 1 to “Condition evaluation for the *cine*-sulfonylation”.

Comment 2: Whether other nucleophilic reagents such as oxygen or sulfur nucleophilic reagents can also undergo *cine* substitution?

Our response: As suggested, different benzoates, sulfonates, *tert*-butoxide, and sulfides are tested, no desired *cine*-substitution products were obtained. Interestingly, diphenylphosphine oxide was successfully involved in the *cine*-substitution reaction to give corresponding product (**7a**) in 69% yield.

Comment 3: For the *cine*-amination, have the authors tried other sources of amines

such as pyrazole, or 1,2,3-triazole, and what's the result?

Our response: As suggested, more *N*-containing heterocycles were evaluated. Pyrazole and 1,2,4-triazole underwent desired cine-amination products (**6i** and **6i**) in 66% and 70% yields. 1,2,3-Triazole was also good substrate for this reaction, delivering cine-amination products (**6k** and **6k'**) in overall 64% yield due to the multiple nucleophilic sites.

Reviewer 2: In this manuscript, Shu and coworkers have developed a cine-substitution reaction of alkenyl thianthrenium salts with nucleophiles (Nu = sulfonates, cyanides, amines, amides) to access alkenyl sulfones, alkenyl nitriles, enamines and enamides. The cine-substitution includes cine-sulfonylation, cine-cyanation, cine-amination, and cine-amidation via C-C, C-N, and C-S bond formation under the mild conditions. The thianthrenium salts can be easily prepared from olefins as demonstrated in the references. This protocol can also be expanded to use the alkenes as the starting materials via the intermediacy of thianthrenium salts in one-pot. This method offers an advantage and is interesting to synthetic chemists. Therefore, the manuscript is recommended for publication in Nature Communications.

General response: We thank the reviewer for supporting the publish of this work in *Nature Communications*! As suggested, the comments are fully addressed in the following sections in detail and corresponding details are provided in the revised manuscript and Supplementary Information.

Comment 1: Page 4-5: X-ray for **4j** (CCDC 2241239) is shown. However, from the picture of the structure, it seems that it is **4k** rather than **4j**. Please check it.

Our response: We thank the reviewer for bringing out this issue. As suggested, the compound was corrected to **4k**.

Comment 2: Reference (Org. Lett. 2023, 25, 5324–5328) is suggested to be added for the description of “On the other hand, alkenyl thianthrenium salts²⁹⁻³⁹...” (page 2)

Our response: As suggested, the mentioned reference (Org. Lett. 2023, 25, 5324–5328) is cited as ref. 40.

Comment 3: Page 9: The sentence “...the formal Heck reaction of alkenes. access to

synthetic useful synthons which are difficult to access from readily accessible starting materials” should be revised.

Our response: We thank the reviewer for bringing out this point. As suggested, we have revised this sentence to “...the formal Heck reaction of alkenes, affording synthetic useful synthons which are difficult to access from readily accessible starting materials”.

Comment 4: 4. Is this reaction specific for cine-substitution or selective for cine-substitution? Is there any ipso-substitution product observed particularly when the reaction yield is not high for cine-substitution product?

Our response: We thank the reviewer for bringing out this point. This reaction is highly selective for *cine*-selectivity. No ipso-substitution was observed. Moderate yields for some substrates resulted from decomposition of vinyl substrates to non-polar impurities.

Comment 5: 5. Page 8: Regarding the mechanism, is there any evidence for the proton transfer and elimination as proposed?

Our response: We thank the reviewer for bringing out this point. Deuterated vinyl thianthrenium salt (**2cc**) was submitted to the reaction, desired cine-substitution product with partial deuterium scrambling (**8c**) was observed, supporting the proposed mechanism that proton exchange between intermediates from vinyl thianthrenium salt and surroundings.

Reviewer 3: In this manuscript, the authors report a unified platform for Heck-type functionalization of olefins by thianthrenation. The strategy is simple in process, without the employment of transition metals, and selectively provides products of Heck-type cine-substitution. This protocol offered new ideas for the construction of cine-C-S, cine-C-C and cine-C-N bonds from olefinic C-H bonds, and is a completely new paradigm in olefinic thianthrene salt chemistry. However, I still have many

questions before the article is published. After clearing these confusions, it can be considered for publication in Nature Communications.

General response: We thank the reviewer for supporting the publish of this work in *Nature Communications*! As suggested, the comments are fully addressed in the following sections in detail and corresponding details are provided in the revised manuscript and Supplementary Information.

Comment 1: Are some olefins with electron-withdrawing groups reactive?

Our response: Styrenes with electron-withdrawing groups derived thianthrenium salts are reaction in this reaction, affording desired cine-substituted products (**6a-6m**) in good yields. However, more electron-deficient olefins such as acrylates proved to be unsuccessful, giving ipso-substitution product in 55% yield.

Comment 2: The crystallographic data in Figure 2 appear to be **4k** rather than **4j**.

Our response: We thank the reviewer for bringing out this point. The error has been corrected as suggested.

Comment 3: The yields of all alkenyl thianthrenium salts should be stated - I note that some of the substrates show yields, but not all.

Our response: As suggested, yields of all alkenyl thianthrenium salts are added in revised supplementary information.

Comment 4: The authors should add examples of asymmetric olefins, e.g. one end of **5i** becomes butyl or a longer chain.

Our response: As suggested, examples of unsymmetric olefins are added. 1-Cyclohexyl-1-pentene was successfully applied to this cine-substitution reaction to give **6r** in 73% yield with >15:1 rr.

Comment 5: In the authors' previous work (doi: 10.1039/D1SC06577G), the allylic product was obtained with 85% yield using Cs_2CO_3 as base; the same 82% yield was

maintained using CH₃CN as solvent. However, in this manuscript, using the same experimental conditions, the Heck-type cine-functionalized product (e.g. product **6l**, **6m**...) was obtained. I think the solvent effect is the key to dominating both products. Please explain the difference between these two strategies in the Mechanistic Investigation section, otherwise, it is easy to mislead our large readership.

Our response: We thank the reviewer for bringing out this point. In general, acetonitrile is good for *cine*-substitution, and DCM is good for allylation reaction (doi: 10.1039/D1SC06577G). Besides solvent effect, there are several factors affect the reaction pathways: 1) The nucleophilicity of the nucleophile. 3) The basicity of nucleophile. 3) The acidity of α -proton of **M1** after nucleophilic addition. If the acidity of α -proton in **M1** increased after addition, *cine*-substitution is preferred. We think the outcome is a combined effect of these factors.

Comment 6: The assigned reactions for the intermediates of the reaction are brilliantly designed. But this raises new questions: under the current reaction conditions, C-S+ bond breaking can also occur from the stabilized intermediates (**7a** & **7b**). So is it possible that the C-S+ bond of alkenyl thianthrenium salts also breaks at the beginning of the reaction and generates an ipso-carbon-negative ion; in such a case the nucleophilic reagent would also attack the cine-position more. Perhaps the authors could experiment with a reaction in which a photocatalyst (or a transition-metal catalyst, but this would probably generate products with only ipso-substitution) is used to induce C-S+ bond rupture in the presence of light, generating an alkenyl radical intermediate, and then the structure of products could be observed to corroborate the mechanism recounted in the paper.

Our response: As suggested, we tested the reaction of **2o** or **2q** in the presence of a nucleophile (**1a** or **1aa**), no desired substitution product was obtained.

Moreover, the use of radical scavengers (TEMPO and BHT) showed no significant inhibition on the reaction (Fig. 4b), excluding the single electron process of this reaction.

Based on the results and further mechanistic experiments (Fig. 4), we believe the current proposed mechanism is feasible and reasonable.

We thank all the reviewers for insightful comments and suggestions!

REVIEWERS' COMMENTS

Reviewer #2 (Remarks to the Author):

My concerns have been addressed in the revisions, therefore the manuscript is recommended for publication.

Reviewer #3 (Remarks to the Author):

The author responded well to our issues and it is now ready for publication.